# Blockchain enabled privacy provisioning scheme for location based services in VANETs

**Iqra Ilyas[1], M. Usman Ashraf [2]\*, Sami S. Albouq[3], Khlood Shinan[4], Hanan E. Alhazmi[5,6], Fatmah Alanazi[7], Saqib Ali[8]**

**1**, **2** Department of Computer Science, GC Women University Sialkot, Pakistan, **3** Faculty of Computer and Information Systems, Islamic University of Madinah, Madinah, Saudi Arabia, **4** Department of Computers, College of Engineering and Computers in Al-Lith, Umm Al-Qura University, Makkah, Saudi Arabia, **5** Computer Science Department, College of Computer and Information Systems, Umm Al-Qura University, Makkah, Saudi Arabia, **6** Department of Cybersecurity, College of Computing, Umm Al-Qura University, Makkah, Saudi Arabia, **7** Computer Science Department, College of Computer and Information Sciences, Imam Muhammad Bin Saud University, Riyadh, Saudi Arabia, **8** Department of Computer Science, University of Agriculture Faisalabad, Pakistan

\* usman.ashraf@gcwus.edu.pk

## Abstract

In recent years, vehicular ad hoc networks (VANETs) have emerged as a crucial component of intelligent traffic systems, offering enhanced road safety through autonomous, distributed, and dynamically structured communication. However, ensuring secure and privacy-preserving message broadcasting in VANETs remains a significant challenge due to their open-access nature. Existing solutions have addressed various security and privacy concerns, yet critical issues such as resistance to traffic analysis, unlinkability of messages, computational efficiency, and location privacy remain underexplored. To bridge these gaps, we propose a blockchain-based privacy-preserving scheme that strengthens VANET security while addressing unobservability, unlinkability, and efficiency in authentication. Our approach leverages a cache-based anonymizer server positioned between the On-Board Unit (OBU) and the Roadside Unit (RSU), which enhances privacy by masking communication patterns and improves efficiency by reducing authentication overhead. Performance evaluations demonstrate that our scheme significantly reduces computational costs, achieving 95.17% to 97.00% reduction in V2V and 97.81% to 98.90% reduction in V2RSU communication time compared to referenced schemes. Additionally, our approach reduces communication cost by 67.94% to 81.67% for V2V and 72.40% to 88.00% for V2RSU, while the location leakage probability is minimized to 0.05% which is significantly lower than centralized architectures. Furthermore, our scheme ensures strong privacy protection, attaining a maximum entropy level of 5 which is 95.8% higher than existing schemes. These results confirm that our framework minimizes computational overhead, optimizes communication efficiency, and enhances privacy protection, making it a robust and scalable solution for VANET systems.

**Data availability statement:** All relevant data are within the manuscript and its Supporting Information files which are uploaded and available at repository: https://figshare.com/articles/online_resource/VANET_privacy_through_BC/28416692

**Funding:** The author(s) received no specific funding for this work.

**Competing interests:** The authors have declared that no competing interests exist.

## 1. Introduction

Vehicle ad-hoc networks (VANETs) play a crucial role in Location-Based Services (LBS) systems, offering a myriad of significant advantages. The integration of VANETs with LBS systems revolutionizes the way we perceive and utilize location-based information. Firstly, VANETs provide real-time and accurate positioning data, leveraging the interconnectedness of vehicles and roadside infrastructure to constantly update and refine location information. This accuracy is paramount in applications such as navigation systems, emergency services, and traffic management, where precise location data is critical for decision-making. Moreover, VANETs enhance the efficiency of LBS systems by enabling dynamic routing and traffic optimization based on current traffic conditions, thereby reducing congestion and travel time. Additionally, VANETs facilitate the dissemination of location-specific information to vehicles in the vicinity, such as nearby amenities, points of interest, and potential hazards, enhancing the overall driving experience and safety [1]. Furthermore, the seamless integration of VANETs with LBS systems opens up opportunities for innovative services, such as collaborative driving, autonomous vehicle coordination, and personalized location-based advertising. In essence, VANETs serve as the backbone of modern LBS systems, empowering them with unprecedented accuracy, efficiency, and functionality, ultimately revolutionizing the way we interact with our environment on the road. For instance, a vehicle equipped with VANET technology is utilizing a navigation system integrated with LBS capabilities. In this scenario, the VANET-enabled LBS system can provide real-time traffic updates and route suggestions based on current road conditions and user preferences [2]. However, the collection and processing of location data in VANETs raise concerns regarding user privacy disclosure. As vehicles transmit their location and trajectory information to the VANET infrastructure and other vehicles in the vicinity, there's a potential risk of unauthorized access or misuse of this data, leading to privacy breaches. To address this issue, VANET-enabled LBS systems must implement robust privacy protection mechanisms. These mechanisms could include encryption of location data during transmission, anonymization techniques to dissociate location information from specific vehicles, and access control mechanisms to restrict data usage to authorized entities only [3]. Moreover, transparent privacy policies should be communicated to users, detailing how their location data will be collected, processed, and shared within the VANET ecosystem. Users should have the option to control the level of granularity of location sharing and to opt-out of certain data collection practices if they so choose [4].

To overcome the privacy challenges in above scenario, blockchain technology holds significant promise in protecting user privacy within VANET Location-Based Service (LBS) systems [5–8]. By its nature, blockchain offers a decentralized and immutable ledger where transactions, in this case, location data exchanges, can be securely recorded and verified without the need for intermediaries. In a VANET LBS system, blockchain can ensure the integrity and confidentiality of location data by encrypting and storing it in tamper-proof blocks across the network. Therefore, we

have proposed a novel Blockchain based privacy protecting approach for VANET system. Our contribution can be summarized as follows:

- To overcome the challenges in centralized systems, this study introduces blockchain technology and devises an innovative decentralized framework known as the blockchain-based VANET. The proposed approach not only guarantees the integrity and security of SBMs but also conserves storage space and reduces processing time within the blockchain through several optimizations. Specifically, we employ a lightweight cryptographic authentication mechanism that minimizes computational overhead. Furthermore, we integrate batch processing with Merkle tree-based validation, allowing multiple authentication requests to be processed together, thereby reducing redundant cryptographic operations and improving efficiency. Further for storage optimization, we adopt ephemeral pseudonyms for key management, which decreases the frequency of blockchain transactions, reducing both storage and processing load.

- To address performance issues in existing distributed systems, we implement cache based anonymizer servers in our scheme to increase the efficiency of the system which is primary concern once a vehicle is connected to LBS system and waiting for query response. This method reduces the computational workload and minimizes the need for direct communication among vehicles, thus improving the privacy and security of vehicle data.

- To further enhance the security of our proposed blockchain-based VANET system, we implement a third security layer in the form of a Certification Authority (CA) which is responsible to authenticate and regulate access to the network, ensuring that only legitimate and authorized entities such as vehicles, roadside units (RSUs), and other network participants are permitted to communicate within the system. The CA issues digital certificates that serve as cryptographic credentials for each network entity, verifying their authenticity before granting access to system resources.

- We evaluate our proposed scheme using the open-source OPNET and Veins vehicular network simulation framework, which serves as a robust platform for validating its effectiveness. Our results demonstrate significant improvements over existing methods in key performance metrics. Specifically, our scheme achieves a 95.17% to 97.00% reduction in V2V computation cost and a 97.81% to 98.90% reduction in V2RSU computation cost, significantly lowering processing overhead. Additionally, we observe a 67.94% to 81.67% reduction in V2V communication cost and a 72.40% to 88.00% reduction in V2RSU communication cost, making our approach more bandwidth-efficient. The integration of a cache-based anonymizer server further enhances system efficiency, achieving a high cache hit ratio. Furthermore, our privacy-preserving measures effectively reduce the probability of location leakage to just 0.05% while ensuring a maximum entropy of 95.8%, demonstrating strong privacy protection. These results confirm the robustness, efficiency, and security of our framework, making it a scalable and effective solution for VANET systems.

Rest of the paper is organized in such way that section 2 describes the existing privacy provisioning state of the art approaches used in VANET systems. Further, section 3 presents the preliminaries and related technologies used in this research. Section 4 demonstrates the problem definition and system architecture. In section 5, we conduct the security analysis of proposed scheme, and present multiple lemmas against possible privacy attacks. Section 6 explain the experimental setup, results and comprehensive discussion. Lastly, we conclude our study in section conclusion section and present all used references at the end of the paper.

## 2. Related work

The issue of privacy arises when sensitive traffic-related messages are vulnerable to unauthorized access and require safeguarding against misuse or exposure. In the realm of vehicular communication, it is imperative to address privacy concerns across all stages of vehicle interaction, encompassing aggregation, processing, collection, evaluation, and visualization. Due to the sensitive nature of the information exchanged, preserving privacy is of utmost importance in this

domain. Privacy preservation has garnered significant attention in recent VANET research endeavors. Scholars have delineated privacy protection into three primary categories: location privacy, trajectory privacy, and identity privacy. Extensive research has delved into this subject, and the most relevant studies are delineated below.

In terms of protective mechanisms, privacy preservation typically falls into two main types: anonymity-based [5] and encryption-based [3] approaches. Anonymity-based methods frequently employ k-anonymity mechanisms, originally devised to safeguard users' location privacy in LBS applications but now widely applied in VANET. By leveraging the principle of maximum entropy, researchers identify k appropriate vehicles with the closest historical request probability, concealing the real vehicle among them to protect vehicular privacy. On the other hand, encryption-based techniques are commonly utilized in VANET authentication, a crucial aspect ensuring the legitimacy of communications. Currently, VANET authentication predominantly adopts asymmetric encryption authentication methods such as Public Key Infrastructure (PKI) [9] and Elliptic Curve Digital Signature Algorithm (EDSA) [10], alongside symmetric encryption authentication mechanisms like group signature authentication [11].

For instance, one study [12] explored the distribution of vehicular pseudonyms utilizing fog computing technology, effectively managed by the edge resources of VANET to enhance location privacy. Similarly, another research effort [13] introduced a policy for trajectory privacy employing multiple mix zones, ensuring the unlinkability of pseudonyms and safeguarding vehicular trajectory privacy.

Moving on to the intersection of blockchain and VANET, blockchain technology has evolved through three generations, from its focus on digital currency transactions (Blockchain 1.0) to the registration, validation, and transfer of smart contracts (Blockchain 2.0), and finally transcending into broader applications encompassing various sectors (Blockchain 3.0). In recent years, researchers have introduced blockchain into VANET, capitalizing on its decentralized, redundant storage, collectively maintained, and tamper-proof characteristics. For instance, Joy. K, and Gerla. M. [14], proposed the concept of blocktree, where vehicles embed their signatures into the blockchain. Dorri, A., Kanhere, S. S., & Jurdak, R. [15], suggested a lightweight scalable blockchain for VANET, and based on this, proposed a blockchain architecture with decentralized privacy protection for intelligent vehicle systems. Moreover, Lei et al. [16] introduced a novel network topology based on blockchain structure to simplify distributed key management in heterogeneous vehicle communication systems. Additionally, blockchain has been proposed as a means for privacy-preserving proof of location in certain contexts [17].

Payattukalanirappel et al. [18] introduced a decentralized authenticated key agreement mechanism utilizing smart contracts to enhance the security of vehicular ad-hoc networks, constituting a VANETs authenticated key agreement scheme based on smart contracts. This scheme operates on a public Blockchain, eliminating the need for a Trusted Authority (TA) for key generation. Within this framework, Roadside Units (RSUs) form the Blockchain network, and smart contracts are executed on the Blockchain. The key agreement protocol incorporates the use of Bloom filters to verify whether public keys are generated by registered vehicles. However, the utilization of Bloom filters in this protocol introduces additional latency. The protocol involves the creation of four Blockchain transactions: one for vehicle registration, another for registering the vehicle's public key, a third for verifying the legality of the roadside unit, and the final transaction for registering the public key of the RSU. Despite these measures, the authentication and key agreement phase is susceptible to man-in-the-middle attacks, as any third party can intercept messages from the sender, fabricate them, and relay them to the receiver.

Similarly, Shen et al. [19] introduced a batch authentication protocol for the Internet of Vehicles, leveraging Blockchain technology. The system comprises several components including Trusted Authority, Road Side Unit, Vehicle, Fog Server, and Cloud Server. The protocol encompasses vehicle-to-vehicle authentication as well as batch authentication, specifically vehicle-to-RSU authentication, where a group of vehicles is authenticated under a particular RSU. The Trusted Authority (TA) is responsible for delivering certificates containing identity, public key, and private key to all vehicles and RSUs. Upon receiving a partial block from the RSU, which includes a list of transactions and their compact signature, the fog node verifies the validity of the signature. If the signature is valid, the partial block is forwarded to the cloud server.

Subsequently, the cloud server converts the partial block into a complete block and conducts mining on the block using the practical Byzantine Fault Tolerance (pBFT) consensus algorithm. However, the addition of the Cloud Server adds complexity to the system and consequently increases the overall latency of the system.

Tandon et al. [20] proposed a specialized vehicle-to-infrastructure handover authentication protocol designed for VANETs, leveraging Blockchain technology. The system incorporates various components, including the Trusted Authority (TA), Road Side Unit (RSU), vehicles, and Blockchain. In the initial registration phase, the TA deploys RSUs and issues smart cards to vehicles, with RSUs responsible for authenticating vehicles before any communication takes place. The Blockchain framework encompasses both the TA and RSUs. Following registration, the TA uploads data pertaining to the pseudo-identity of registered vehicles onto the Blockchain. Throughout the authentication process, RSUs verify the registration status of vehicles. Upon successful completion of vehicle-to-infrastructure authentication, RSUs initiate a transaction on the Blockchain, comprising the vehicle's temporal identity, a randomly generated number, and the RSU's signature. It is worth noting that while the paper delves into vehicle-to-infrastructure communication, it fails to address the critical aspect of V2V communication within the VANET framework.

Weng et al., [21] introduced TCEMD scheme that employs a trust-cascading mechanism where vehicles dynamically assess and propagate emergency messages based on trust values derived from historical interactions and behavioral consistency. The model integrates a multi-layer trust evaluation process, combining direct trust (from past interactions) and indirect trust (from recommendations) to ensure only credible messages are disseminated, reducing the risk of false alarms. By prioritizing highly trusted vehicles for message forwarding, TCEMD enhances road safety, minimizes redundant transmissions, and accelerates emergency response times. However, the reliance on historical data may limit the model's adaptability to newly joined vehicles. Additionally, malicious nodes could still exploit indirect trust mechanisms to manipulate message dissemination. Computational overhead in trust evaluation may also impact real-time performance in high-density traffic scenarios. Similarly, Guo et al., [22] worked on VANETs safety and privacy for end users and proposed TROVE scheme where context awareness based RL mechanism was used to improve the road safety. The proposed TROVE dynamically assesses vehicle trustworthiness based on contextual information, such as mobility patterns, message consistency, and behavioral history, rather than relying solely on predefined rules. The model employs RL to continuously learn and adapt trust values by rewarding reliable interactions and penalizing malicious behavior, ensuring robust and adaptive decision-making. This approach enhances security, mitigates false data injection attacks, and improves trust accuracy in VANET communications. However, reliance on historical data may lead to delayed responses in rapidly changing traffic scenarios. Additionally, the computational complexity of reinforcement learning could pose scalability challenges. Furthermore, adversarial RL techniques might exploit vulnerabilities in the trust model, leading to manipulated trust scores.

According to Liu et al., [23] intelligent transportation systems can significantly enhance driving safety and optimize traffic flow, particularly with advanced AI models. Leveraging cloud technology improves their efficiency, giving rise to cloud-integrated vehicular networks. Trust assessment is vital for secure communication, but updating reputation metrics places a heavy load on the Central Authority (CA). Authors proposed a Secure Reputation Enhancement (SRE) framework based on ECC and Paillier encryption, shifting computational tasks to a semi-trusted Cloud Service Provider (CSP). The solution cuts CA's workload by 88.36% in processing and 83.88% in data transmission while provisioning privacy and security standards. Extensive evaluations demonstrate it's clear advantages over conventional methods. However, reliance on a semi-trusted CSP may introduce security risks if compromised. Additionally, potential latency issues in cloud processing could affect real-time decision-making in vehicular networks.

The literature analysis outlined above highlights the prevalent challenges faced by existing systems, notably concerning substantial computational and communication burdens. Achieving an optimal equilibrium between communication and computation costs remains a rarity. Moreover, some systems depend on multiple trusted third parties, demanding trust from both users and infrastructure, resulting in heightened communication overhead and latency. Additionally, reliance

on centralized databases in certain systems introduces transparency deficiencies and single points of failure risks. The proposed protocol seeks to address and alleviate these identified issues.

## 3. Preliminaries

### A. Blockchain technologies

Blockchain technology stands as a pivotal emerging technology with far-reaching implications across various sectors. Its significance lies in its ability to revolutionize traditional systems by providing decentralized, transparent, and immutable records of transactions. One of its primary applications is in finance, where it enables secure and efficient peer-to-peer transactions without the need for intermediaries. Beyond finance, blockchain finds utility in supply chain management, ensuring transparency and traceability of goods from production to consumption. Additionally, its decentralized nature holds promise for enhancing data security and privacy in healthcare, voting systems, and identity management [24]. Moreover, blockchain facilitates the creation of decentralized applications (DApps) and smart contracts, automating processes and reducing reliance on centralized authorities. As blockchain technology continues to evolve, its potential to disrupt existing paradigms and foster innovation across diverse industries solidifies its status as a transformative force in the digital era. Blockchain technology, denoted by $B$, is increasingly utilized for privacy protection in various domains due to its inherent characteristics such as decentralization, immutability, and transparency. In a blockchain-based privacy protection scheme, each participant, denoted by $P_i$, possesses a unique cryptographic key pair ($SK_i, PK_i$) where $SK_i$ represents the private key and $PK_i$ denotes the public key. Transactions, denoted by $T$, are securely recorded on the blockchain using cryptographic hashing functions, ensuring data integrity and preventing tampering. Privacy protection is achieved through the use of pseudonyms, denoted by $Pseudo_i$, which are derived from the public key $PK_i$ and used to conceal the true identity of participants. Smart contracts, denoted by $SC$, facilitate the enforcement of privacy policies and consent management, dictating how data is accessed and utilized. Moreover, zero-knowledge proofs, denoted by $ZKP$, enable participants to prove the validity of a statement without revealing any sensitive information. Through the integration of these cryptographic techniques and decentralized consensus mechanisms such as Proof of Work (PoW) or Proof of Stake (PoS), blockchain technology ensures robust privacy protection while preserving data transparency and integrity in a trustless environment. The basic architecture of blockchain is presented in Fig 1 as follows.

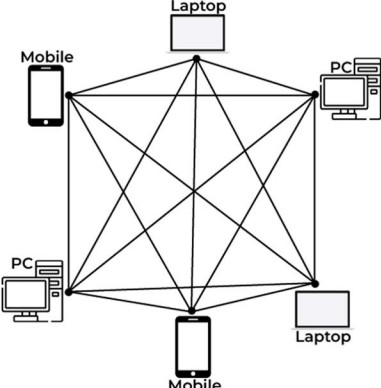

**Fig 1. Fundamental architecture of blockchain.**

## B. K-Anonymity Approach

K-anonymity is a fundamental concept in privacy-preserving data anonymization, aiming to mitigate the risk of individual re-identification in datasets containing sensitive information. It provides a robust framework for privacy protection in data-sets containing sensitive information. By ensuring that each individual's data is sufficiently anonymized and grouped with similar records, k-anonymity helps to protect against privacy breaches and unauthorized disclosure of personal information [25]. At its core, k-anonymity ensures that each data record remains indistinguishable from at least $k-1$ other records with respect to certain attributes. This means that for any given data record $di$ in a dataset $D$, there exists at least one other record $dj$ sharing the same sensitive attributes $S$ as $di$, where the total number of such records $|dj|$ is equal to or greater than $k$. Mathematically, this concept is expressed in equation 1 as follows:

$$\forall di \in D, \exists dj \in D \text{ s.t. } di[S] = dj[S] \tag{1}$$

Where $|dj| \geq k$. To put it simply, k-anonymity ensures that individual records in a dataset are grouped together with enough similar records to obscure the identity of any specific individual. This makes it significantly more challenging for adversaries to pinpoint and re-identify individuals based on sensitive attributes alone. Achieving k-anonymity involves applying various anonymization techniques to the dataset. Generalization involves replacing specific attribute values with more generalized ones, such as replacing exact ages with age ranges. Suppression, on the other hand, entails removing certain attributes entirely from the dataset to prevent identification. Perturbation techniques add noise or randomization to the data, making it more difficult for adversaries to infer specific details about individuals.

## C. Dynamic threshold encryption (DTE)

Dynamic threshold encryption (DTE) is a cryptographic scheme that allows for the secure sharing of a secret among a group of participants, with the ability to dynamically adjust the threshold for decryption. It provides a flexible and adaptable approach to secure data sharing among multiple participants by allowing the threshold for decryption to be dynamically changed as needed [26]. In DTE, a secret $S$ is encrypted using a public key derived from shares distributed among $n$ participants. Unlike traditional threshold encryption schemes where a fixed number of shares are required for decryption, DTE enables the threshold value $t$ to be dynamically changed based on certain conditions or criteria. DTE consist of four primary steps discussed as follows:

**Key Generation:.**

- *Each participant i generates a pair of public-private keys (PKi,SKi).*

- *Each participant i calculates their share $Share_i$ using their private key $SK_i$.*

**Encryption.**

- The secret S is encrypted using a public key PK derived from the shares: E(S)=Encrypt(S,PK)

**Decryption:.**

- *To decrypt the encrypted secret E(S), at least t participants need to collaborate.*

- *Suppose T is the set of indices of the participants who collaborate for decryption, where $|T| \geq t$.*

- *The combined private keys of the collaborating participants are used to recover the secret: $S = Decrypt(E(S), \{SK_i \mid i \in T\})$*

**Dynamic Threshold Adjustment:.**

- *The threshold value t can be dynamically adjusted based on specific conditions or events.*

- *For example, if the number of active participant's changes, the threshold value can be recalculated to ensure the required level of security.*

## 4. Problem definition and system architecture

This section commences with an overview of our system structure followed by the problem definition. Next, we introduce the extended blockchain tailored for VANETs. Subsequently, we describe a succinct overview of registration process along with authentication procedure employed in the proposed trust management model. Finally, we delineate several fundamental assumptions crucial for ensuring the seamless operation of the system.

### A. Problem definition

In VANET systems, ensuring privacy for vehicular communication is of paramount importance to prevent unauthorized access to sensitive information and protect user anonymity. Let $G=(V,E)$ denote a VANET network, where $V$ represents the set of vehicles and $E$ represents the set of communication links between vehicles. Each vehicle $vi$ in $V$ is associated with a set of attributes denoted by $Attr(vi)$, including location, speed, direction, and identity information. The goal is to design privacy provisioning mechanisms that preserve the privacy of vehicle attributes while allowing for efficient and secure communication within the network.

Several privacy-preserving mechanisms have been proposed to enhance security and anonymity in VANETs. One widely used approach is pseudonym-based authentication [27], where vehicles periodically change their pseudonyms to prevent long-term tracking by adversaries. While this technique improves privacy, it introduces a high communication overhead, as frequent pseudonym updates require continuous coordination with certificate authorities. Another method, group signature schemes [28], allows vehicles to authenticate messages anonymously within a group, ensuring privacy. However, these schemes suffer from complex key management and high computational costs, making them less suitable for large-scale networks with real-time communication requirements. Additionally, mix networks [29] and $k$-anonymity techniques have been explored to obscure communication patterns and enhance anonymity. Mix networks achieve this by routing messages through multiple relay nodes, while $k$-anonymity ensures that each vehicle is indistinguishable from at least $k-1$ other vehicles. Although these approaches provide strong privacy guarantees, they introduce significant latency and rely on trusted third parties, limiting their practicality in dynamic VANET environments. More recently, blockchain based privacy solutions have gained attention due to their decentralized architecture and robust security features. Blockchain enables tamper-proof identity management and secure message verification; however, frequent transactions in blockchain-based VANET systems result in high storage and processing requirements, which can overwhelm resource-constrained vehicular networks. Despite these advancements, existing privacy-preserving solutions still exhibit critical limitations. Many approaches impose high computational and communication overhead, making real-time processing inefficient. Scalability remains a challenge, as the increasing number of vehicles and messages leads to network congestion and excessive resource consumption. Furthermore, existing authentication mechanisms often require complex cryptographic operations, making them impractical for real-time vehicular communication. Addressing these limitations requires an efficient, scalable, and lightweight privacy-preserving framework that enhances security while minimizing resource consumption in VANETs.

### B. System architecture

To protect vehicle's identity and spatial information, we propose a blockchain based VANET system as shown in Fig 2. The system illustrates the network model that contain different components including vehicles, OBUs, anonymizer server, RSUs, core network (that further include CA, LBSP, CSP), blockchain (BC), and blockchain network (BCN).

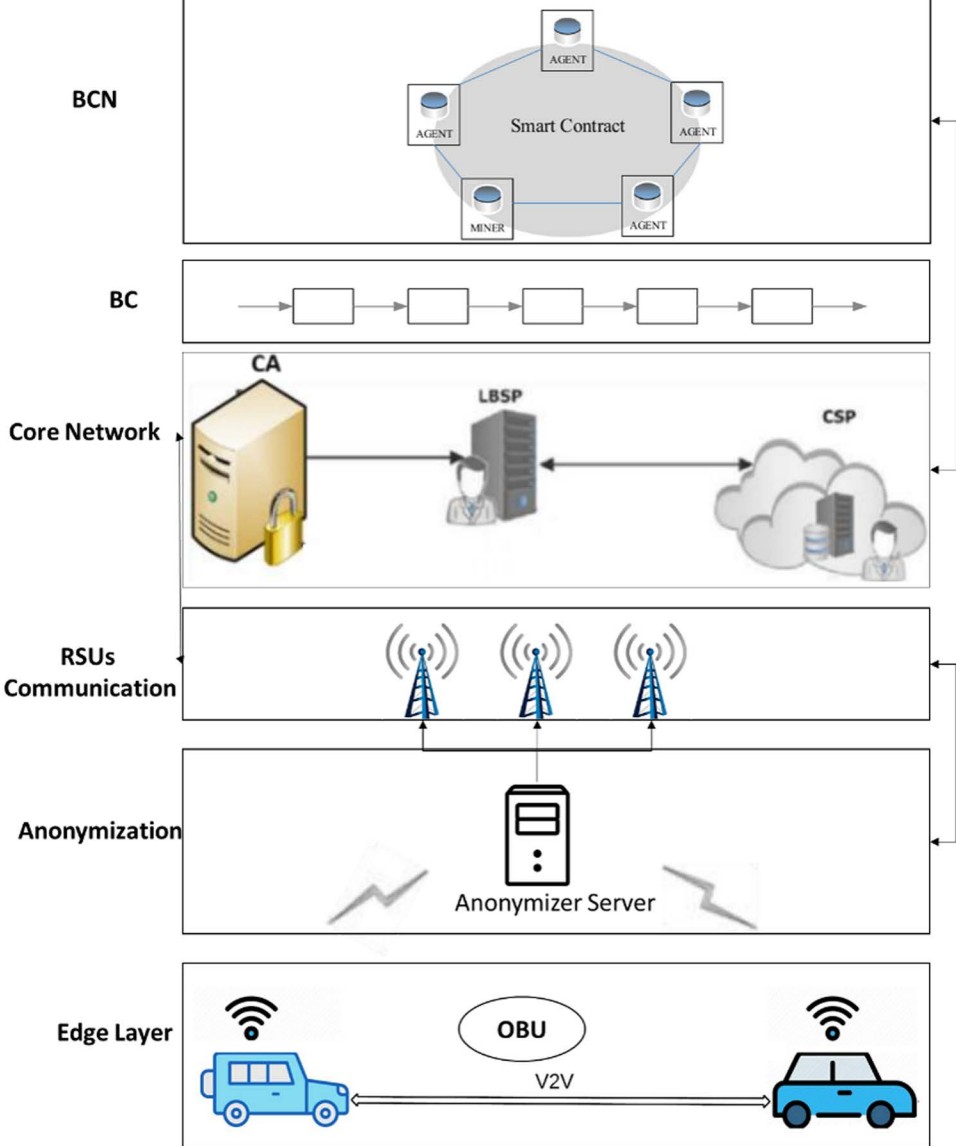

**Fig 2. Block diagram of proposed privacy provisioning for VANETs.**

**Mobile entities (vehicles).** In a Vehicular Ad-Hoc Network (VANET) system, vehicles play a pivotal role as mobile nodes that dynamically form a communication network. Vehicles are represented as a set $V=\{v1,v2,...,vn\}$, where $n$ denotes the total number of vehicles. Each vehicle $vi$ is equipped with an On-Board Unit (OBU), denoted as $OBUi$, responsible for facilitating communication within the network. Vehicles play a crucial role in exchanging information, such as Safety Beacon Messages (SBMs), with other vehicles and Road Side Units (RSUs) to enhance safety and efficiency on the road. The communication capabilities of vehicles are essential for enabling various VANET applications, including collision avoidance, traffic management, and emergency assistance. Mathematically, the presence of vehicles in the VANET system can be expressed as $V=\{v1,v2,...,vn\}$, where each vehicle is equipped with $OBUi$ to participate in communication protocols and data exchange.

**On board unit.** In the VANET system, the On-Board Unit (OBU) serves as a critical component for each vehicle $v_i$. Mathematically, we represent the set of OBUs as $O=\{OBU1, OBU2,...,OBUn\}$, where $n$ denotes the total number of vehicles in the network. The OBU $OBUi$ associated with each vehicle facilitates communication and data exchange within the VANET. It enables vehicles to transmit and receive Safety Beacon Messages (SBMs) and other relevant information to enhance road safety and efficiency. The OBU is responsible for implementing communication protocols, such as Vehicle-to-Vehicle (V2V) and Vehicle-to-Infrastructure (V2I) communication, and ensuring secure and reliable transmission of data. Mathematically, the presence of OBUs in the VANET system can be expressed as $O=\{OBU1, OBU2,...,OBUn\}$, where each OBU $OBUi$ is essential for enabling vehicle communication and supporting VANET functionalities.

---

**Algorithm 1: On-Board Unit Operations**

**Input:**
>    *query*: Query or message from the vehicle
>    *cachedData*: Cached data stored in the OBU

**Output:**
>    *Qresponse*: Response to the query or message

**Begin**
  1. **If** *query* is not empty, **then**
  2.    **If** *query* exists in *cachedData*, **then**
  3.        Retrieve response from cache: response ← getCachedResponse(*query*)
  4.        **Return** response
  5.    **Else**
  6.        Forward query to the Anonymizer Server: response ← sendToAnonymizer(*query*)
  7.        Cache the response for future queries: cacheResponse(query, response)
  8.    **Return** response

**End**

---

**Anonymizer Server.** In term of privacy provisioning, the Anonymizer Server plays a vital role in preserving privacy by anonymizing the identity and location information of vehicles. Mathematically, we denote the Anonymizer Server as $A$. It is linked with each On-Board Unit (OBU) $OBUi$ associated with vehicles in the network. The link between the Anonymizer Server and the OBU is represented as a function $O \rightarrow A$, where $O$ is the set of OBUs and $A$ is the Anonymizer Server. The Anonymizer Server receives sensitive information, such as location data and vehicle identifiers, from the OBUs. It then applies anonymization techniques to obfuscate this information before forwarding it to other entities in the network. By anonymizing data at the source, the Anonymizer Server helps protect the privacy of vehicles and their occupants while enabling efficient communication within the VANET system. Mathematically, the link between the OBUs and the Anonymizer Server can be expressed as $f:O \rightarrow A$ indicating the association between each OBU and the Anonymizer Server for privacy protection purposes.

---

**Algorithm 2: Anonymization Operations**

**Inputs:**
>    vehicleData: Vehicle data to be anonymized
>    cloakingRegion: Cloaking region information

**Output:**
>    anonymizedData: Anonymized data

**Begin:**
  1. **If** *vehicleData* is not empty, **then**
  2.    **If** *cloakingRegion* is provided, **then**
  3.        **If** *vehicleData* is valid within *cloakingRegion*, **then**
  4.            Anonymize vehicle data: *anonymizedData* ← anonymize(*vehicleData, cloakingRegion*)
  5.    **Return** anonymizedData

**End**

---

**Road Side Units (RSUs).** Road Side Units (RSUs) play a crucial role in facilitating communication and providing infrastructure support for vehicles. Mathematically, we denote the set of RSUs as $R=\{RSU1, RSU2,...,RSUm\}$, where

*m* represents the total number of RSUs deployed in the network. Each RSU *RSUi* is strategically positioned along roadsides to cover specific geographical areas. RSUs serve as communication hubs, enabling Vehicle-to-Infrastructure (V2I) communication and enhancing connectivity within the VANET. They relay information between vehicles and the infrastructure, such as traffic updates, road conditions, and safety alerts. Mathematically, the presence of RSUs in the VANET system can be expressed as *R*={*RSU*1,*RSU*2,...,*RSUm*}, where each RSU *RSUi* contributes to the network's functionality by providing infrastructure support and enabling communication between vehicles and the roadside infrastructure. Algorithm 3 presents the registration process of RSUs.

---

**Algorithm 3: Registration process of RSUs in VANET**

**Inputs**:
 rsuID: ID of the RSU
**Output**:
 bool: Boolean indicating the success of the registration
**Begin**:
1. **If** *msgSender* is not null, **then**
2. **If** *rsuAddress* is not equal to *msgSender*, **then**
3. **If** *rsuID* is not empty, **then**
4. Set registration flag: *isRegd*←False
5. Assign *msgSender* as *rsuAddress*
6. Map *rsuID* to *msgSender*: RSUMapping[*msgSender*].id←*rsuID*
7. Mark *msgSender* as registered: RSUMapping[*msgSender*].isRegd←True
8. **Return** bool indicating successful registration
**End**

---

**Core Network Layer.** In the VANET system, the core network layer comprising the Certificate Authority (CA), Location-Based Service Provider (LBSP), and Cloud Service Provider (CSP) plays a major role in managing and processing data received from Roadside Units (RSUs) before forwarding it to the blockchain layer. We denote the components of the core network layer as *C*= {*CA*, *LBSP*,*CSP*}. Upon receiving data from RSUs, the CA, as a trusted entity, verifies the authenticity of the data and issues digital certificates to ensure secure communication within the network. The LBSP, responsible for location-based services, processes location data received from RSUs to provide relevant services to vehicles, such as navigation assistance and traffic updates. The CSP, serving as a cloud-based platform, stores and manages data received from RSUs, ensuring scalability and accessibility. Mathematically, the functions of the core network layer can be expressed as *C*= {*CA*, *LBSP*,*CSP*}, where each component contributes to data processing, verification, and management before forwarding it to the blockchain layer for further processing and storage.

**Blockchain (BC), and Blockchain Network (BCN).** In the VANET system, Blockchain (BC) serves as a decentralized and immutable ledger to protect the privacy of vehicles by securely storing and managing sensitive information. Mathematically, we denote the Blockchain as *BC*, which is further linked with the Blockchain network (BCN). The Blockchain network consists of a distributed network of nodes responsible for validating and storing transactions. The role of BC in VANET is crucial for ensuring the integrity and privacy of data exchanged between vehicles and infrastructure. BC utilizes cryptographic techniques, such as hashing and digital signatures, to secure transactions and prevent unauthorized access. By linking with the BCN, BC ensures transparency, decentralization, and tamper-resistance, making it an ideal platform for preserving the privacy of vehicle data. Mathematically, the association between BC and BCN can be represented as *BC*→*BCN*, indicating the linkage between the Blockchain and its network for protecting the privacy of vehicle information in the VANET system.

## C. Workflow of Proposed model

Addressing privacy concerns within the blockchain-based VANET system necessitates meticulous control over all participating nodes. Hence, this study advocates the implementation of multiple privacy protection mechanisms

at different stages to ensure the privacy provisioning to the user's sensitive information. For instance, anonymizer server (AS) is deployed at initial stage to anonymize the vehicle's location/identity information once it is forwarded to RSU through OBU. RSU point could be harmful to disclose the sensitive information. Anonymizer servers generate the pseudonym data of received information from OBU and forwarded to RSU unit along with cloaking region. In this way, a secure path is initiated at first stage. In the next stage, the anonymized information is disseminated over the core network where CA generate a certificate against the received query and forward to LBSP module as shown in Fig 3.

Further, our proposed model advocates the implementation of a private chain to construct the blockchain network. Within this framework, the hash values of all data within the core network are securely stored within the blockchain network. This measure serves as protection against tampering and manipulation by malicious entities. Any alteration to the data results in a change in its hash value, readily detectable within the auditable blockchain. Moreover, by storing only the hash values, considerable storage resources within the blockchain network are conserved, consequently enhancing system responsiveness. Smart contracts constitute another integral aspect of the proposed solution. These contracts embody predefined rules and conditions that are automatically executed once triggered. In this context, specific rules pertaining to authentication, k-anonymity, sub-identity generation and dynamics, SBMs recording, and more are encoded into smart contracts. By functioning solely based on code, smart contracts not only reduce transaction costs but also enhance accuracy and efficiency by obviating the need for human intervention. Agent nodes play a pivotal role within the blockchain network as participating entities responsible for maintaining the integrity of transactions through consensus mechanisms. Each agent node contributes to the consensus process and maintains a backup of the blockchain network's data. In this study, agent nodes adopt a Proof of Work (PoW) approach for achieving consensus. Furthermore, within the realm of agent nodes, a subset known as miners holds a distinct position. Miners are agent nodes that successfully solve mathematical problems and thus earn the privilege of maintaining legal blocks within the blockchain. These miners are tasked with processing new blocks and validating data before updating the blockchain network's backup data. As such, miners not only participate in the consensus process but also possess the capability to mine and validate new blocks, rendering them instrumental in the network's operation.

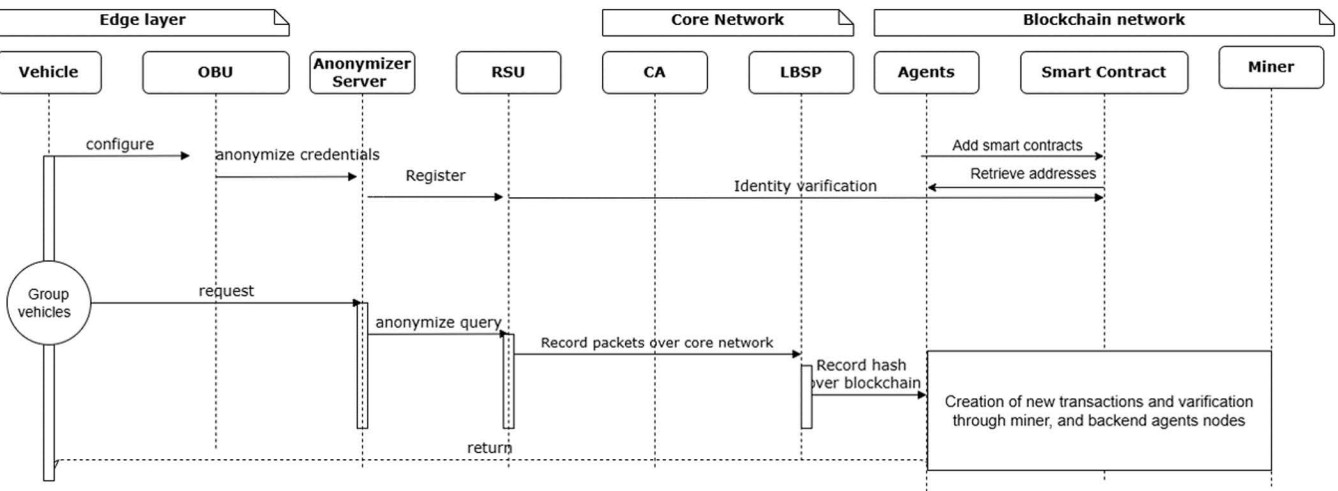

**Fig 3. Workflow order of proposed privacy provisioning for VANETs using sequence diagram.**

## 5. Thread models, and security analysis

This section illustrates that the proposed scheme effectively fulfills the security and privacy criteria outlined for vehicular communication in the design objectives subsection. First we present the threat models; then possible resistance using our proposed scheme.

### Adversary model

In the VANET environment, attackers can be categorized based on their capabilities into passive and active adversaries. Passive adversaries primarily engage in eavesdropping, intercepting communication channels to gather sensitive information without altering the transmitted data. Their goal is to analyze traffic patterns, extract location details, or infer private information about vehicles and drivers. In contrast, active adversaries take a more aggressive approach by injecting, modifying, replaying, or impersonating legitimate messages to disrupt the network's security and privacy. These attackers may manipulate query content, replay outdated messages to create confusion, impersonate authorized vehicles to gain unauthorized access, or launch man-in-the-middle (MITM) attacks to intercept and alter communications between legitimate entities. Given these adversarial threats, the proposed scheme implements robust privacy-preserving mechanisms, ensuring secure communication and protecting VANET operations from potential security breaches.

### Attack vectors and corresponding threats

The following attack vectors outline potential security risks in the proposed VANET system presented in Table 1:

  A.  *Query content modification attack resistance.*  Let $S$ represent the set of legal vehicle-related sensitive information, and let $'S'$ represent the set of pseudonym data forwarded by the anonymizer server. Additionally, let $C$ represent the cloaking region generated by the anonymizer server and forwarded to the RSU. Then, the lemma can be expressed as follows:

  Lemma: For any adversary $A$ attempting to tamper with the legal vehicle-related sensitive information $S$, such that $S \neq S'$, in the presence of the anonymizer server and the proposed scheme, the adversary's ability to modify $S$ is significantly impeded. Mathematically, this can be represented as:

$$\forall S \in S, \ \forall A : \ P\left(A \ modifies \ S \ | \ S \neq S', C\right) \ll P\left(A \ modifies \ S \ | \ S = S', \ C\right) \tag{2}$$

Where $P\left(A \ modifies \ S \ | \ S \neq S', C\right)$ denotes the probability of the adversary $A$ successfully modifying the legal vehicle-related sensitive information $S$ when $S$ is different from the pseudonym data $'S'$ and the cloaking region $C$ is present. $P(A$

**Table 1. Possible threats, their descriptions and mitigation using our proposed scheme.**

| Threat | Attack Description | Mitigation in Proposed Scheme |
|---|---|---|
| Query Content Modification Attack | An attempt to modify query content exchanged between OBUs and RSUs. | Anonymization via pseudonymization ensures that vehicle-sensitive data is masked before reaching RSUs, reducing the feasibility of content modification. |
| Content Replay Attack | Attempt to resend previously captured messages with modified timestamps to mislead the system. | The system verifies timestamps of messages, rejecting any replayed message with an invalid or outdated timestamp. |
| Impersonation Attack | Impersonate legitimate vehicles to gain unauthorized access to services. | Certificate Authority (CA) authenticates vehicles, ensuring only registered entities with valid credentials can communicate. |
| Man-in-the-Middle (MITM) Attack | Intercept and modify communication between legitimate entities (e.g., vehicles and RSUs). | Secure communication channels established through anonymizer servers and blockchain-based integrity verification prevent unauthorized message modification. |

modifies $S \mid S = S'$, $C$) denotes the probability of the adversary $A$ successfully modifying the legal vehicle-related sensitive information $S$ when $S$ matches the pseudonym data $'S'$ and the cloaking region $C$ is present.

**B. Content replay attack resistance.** For any attacker $A$ attempting to perform a replay attack by using a different timestamp $'T'$ in place of the original timestamp $T$, the proposed system effectively detects and rejects the replayed message. This can be represented as:

$$\forall T, \ T' \ \in \ Timestamps, \ \forall A: \ P\,(A \ T') \ \ll \ P\,(A \ T) \tag{3}$$

Where $P\,(A$ successfully replays message with $T')$ denotes the probability of the attacker $A$ successfully replaying a message with a different timestamp $'T'$. $P\,(A$ successfully replays message with $T)$ denotes the probability of the attacker $A$ successfully replaying a message with the original timestamp $T$. This proof indicates that the proposed system effectively mitigates replay attacks by verifying the freshness of timestamps in traffic-related messages. Since using a different timestamp result in a different value for the inequality check, replayed messages are promptly detected and rejected.

**C. *Impersonation attack resistance*.** Let $CA$ represent the Certificate Authority in the core network of the VANET system. The lemma against impersonation attack resistance can be formulated as follows:

Lemma: The proposed VANET system, utilizing a Certificate Authority (CA) for query authentication, effectively resists impersonation attacks. Mathematically, this can be expressed as:

$$\forall attacker \ A, \ P\,(A) \ \ll \ P\,(CA) \tag{4}$$

Where, $P\,(A$ successfully impersonates as a legitimate user) denotes the probability of an attacker $A$ successfully impersonating as a legitimate user without being detected. $P$ (legitimate user successfully authenticates with CA) denotes the probability of a legitimate user successfully authenticating with the Certificate Authority (CA) to obtain valid credentials. This lemma asserts that the use of the Certificate Authority (CA) significantly reduces the likelihood of impersonation attacks in the VANET system. Legitimate users' authentication through the CA adds an extra layer of security, making it challenging for attackers to successfully impersonate legitimate users without valid credentials.

**D. *Middle man attack resistance*.** Let $S$ denote the sender of a message, $V$ denote the verifier, and $M$ represent the message transmitted in the VANET system. The lemma for Resistance to Man-in-the-Middle (MITM) Attack in the described scenario can be formulated as follows:

Lemma: In the VANET system architecture described, incorporating an anonymizer server connected with the On-Board Unit (OBU), Certificate Authority (CA) within the core network containing the Location Based Service Provider (LSP), and cloud service provider (CSP), followed by linkage with blockchain and further blockchain network, the system effectively resists Man-in-the-Middle (MITM) attacks. Mathematically, this can be expressed as:

$$\forall M, \ S, \ V: \ P\,(SV \ relationship \ is \ verified) \ = \ 1 \tag{5}$$

Where $P\,(SV$ relationship is verified) denotes the probability that the relationship between the sender $S$ and the verifier $V$ is verified, ensuring that the message $M$ is transmitted securely without intervention. $M$ represents the message transmitted in the VANET system. It shows that in the described system architecture, the verification of the relationship between the sender and the verifier is a necessary condition for ensuring message validity and authenticity. By establishing this relationship and ensuring that a genuine message cannot be changed or fabricated, the system effectively mitigates the risk of Man-in-the-Middle (MITM) attacks.

## 6. System evaluation, results and discussion

This section presents the effectiveness of our proposed blockchain based privacy provisioning model with respect to vehicle privacy and unity. The initial segment delineates the simulation environment, while the subsequent portion scrutinizes simulation outcomes utilizing four key metrics: computation cost, communication cost, probability of privacy leakage, and entropy. To ascertain the efficacy of our proposed architecture, we juxtapose it with well-known existing state-of-the-art methods including centralized system [10] and decentralized framework [19], and cache based centralized [20]. The primary disparity between the distributed architecture and our proposed framework lies in the data privacy layers, and data handling. In the blockchain network of the distributed architecture, SBMs data is directly stored, whereas in the blockchain network of our proposed architecture, we store the hash values of the SBMs data. Fig 4 describes the deployment of blockchain based smart contract in proposed privacy provisioning method.

To optimize the simulation experiment, we segment our proposed architecture into three primary parts: (1) a cache-based anonymizer server, (2) the VANET network, and (3) the blockchain network. The cache-based anonymizer server serves as the initial layer, interfacing with OBUs to protect the privacy-related information by anonymizing actual data. Additionally, it efficiently handles query responses from the server cache, promptly delivering results when the desired information is accessible. This approach significantly reduces the computational and communication overhead of the entire system. The VANET network primarily involves vehicles uploading SBMs, while the blockchain network is responsible for recording transactions. We utilize OPNET [30] and the Veins framework [31] for simulating the VANET and blockchain networks, respectively.

The Veins framework, being an advanced open-source vehicular network simulation framework, facilitates the implementation of blockchain-based computing platforms, incorporating smart contracts and the PoW consensus mechanism. Table 2 presents the configuration details used in the experiments. In our experiments, we leverage the Veins platform to establish rules for authentication, k-anonymity unity, sub-identity dynamic change rules, and SBMs recording within smart

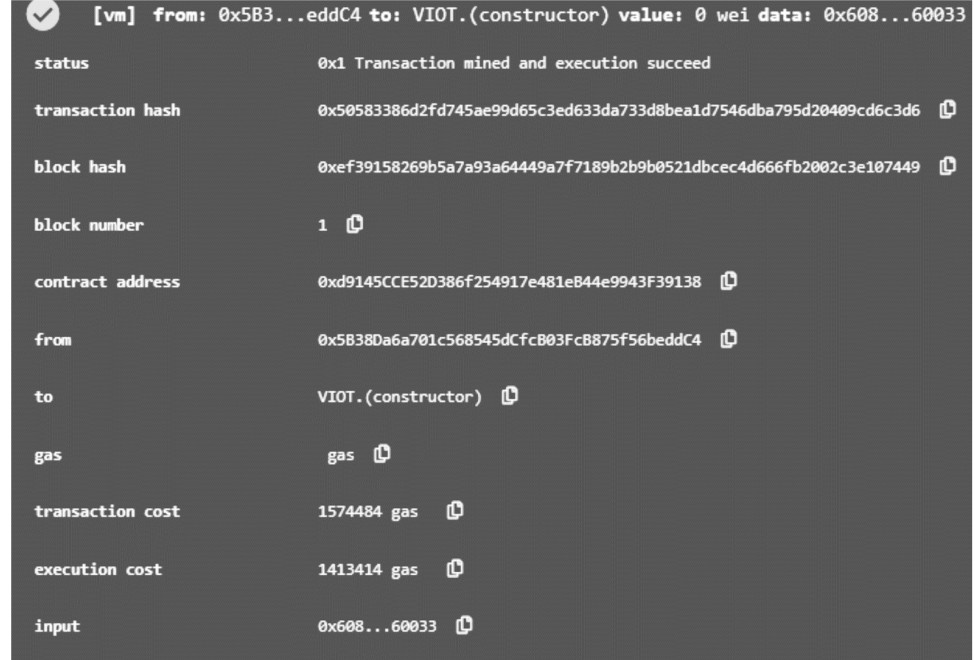

**Fig 4. The deployment of blockchain based smart contract in proposed privacy provisioning method.**

**Table 2. Configuration attributes for experimental setup.**

| Attributes | Values |
|---|---|
| Selected Simulation area | 6 x 6 (KM) |
| Time | 500 Secs |
| No. of Tracing Trusted authority | 1 |
| Key gen center | 1 |
| range of the communication | 500 m |
| RSUs | 10 |
| MAC Protocol | IEEE 802.11p |

contracts. For the blockchain network, we employ the PoW consensus to validate new data blocks. During experimentation, we simulate a target vehicle U navigating a trajectory Tr, comprising 10 locations {l1, l2,…,l10}. To ensure privacy, a corresponding undirected graph is generated for each location, resulting in 10 undirected graphs {G1, G2,…, G5} for vehicle U. We conduct experiments eight times to accurately assess performance, calculating average values for system time, average distance, connectivity, and privacy leakage entropy. Additionally, we simulate two traffic scenarios, light and heavy, with the heavy traffic scenario involving a traffic density of 0.5 v/h (i.e., 5000 vehicles passing through Tr in one hour), while the light traffic scenario involves a density of 0.1 v/h.

**A. Computation cost.** To evaluate our proposed privacy provisioning framework, we evaluate different metrics where computation cost is the first one that refers to the computational resources required to perform various operations such as encryption, decryption, hashing, verification, and other processing tasks. This cost is crucial to consider as it directly impacts the efficiency and performance of the system. Mathematically, the computation cost $C$ can be expressed as the sum of the costs associated with individual operations:

$$C(comp) = \sum_{i=1}^{n} C_i$$

Where, $C(comp)$ is the total computation cost, $n$ is the total number of operations, and $C_i$ represents the cost of each individual operation. The computation cost for different operations can vary depending on factors such as algorithm complexity, key length, message size, and processing power of the devices involved. To calculate the computation cost in VANET systems, we measured the processing time required for key cryptographic operations in different vehicular network scenarios. Specifically, we considered the time taken for encryption and decryption of messages exchanged between vehicles and infrastructure, the hashing process for integrity verification using cryptographic hash functions, and the time required for digital signature generation and verification for authentication purposes. The computation cost analysis is illustrated in Fig 5 as outlined below. In terms of V2V communication, our scheme demonstrates a significantly lower computational cost compared to the referenced schemes, with a value of 0.23972 ms. this indicates a substantial reduction in computational overhead, enhancing efficiency. Similarly, for V2RSU communication, our scheme exhibits a substantially lower computational cost of 0.13166 compared to the referenced schemes, further emphasizing the efficiency gains achieved. These results suggest that our proposed scheme effectively minimizes computational costs, making it a promising solution for VANET systems.

**B. Communication Cost.** In our experiments, the second evaluation metric is communication cost that refers to the resources expended in transmitting data between vehicles, roadside units (RSUs), and other network components. It encompasses factors such as bandwidth usage, message overhead, and transmission delays. Mathematically, the communication cost $C(com)$ is expressed in equation 7 as follows:

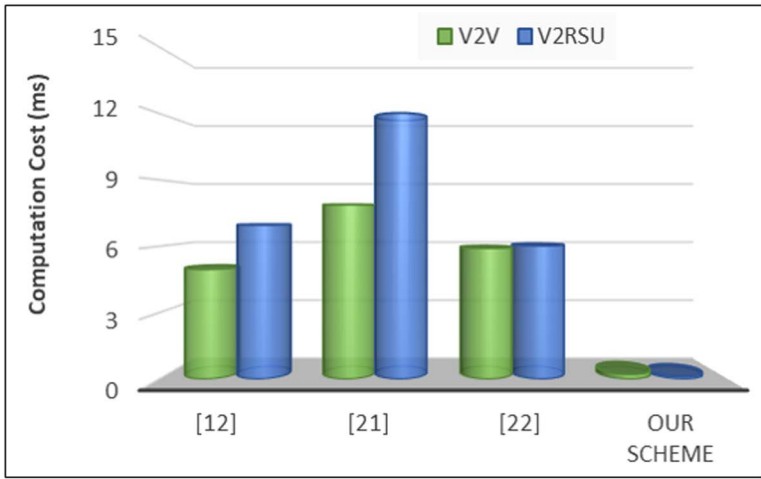

**Fig 5. Computation cost for V2V and V2RSU.**

$$C(com) = \sum_{i=1}^{n} (B_i \times T_i)$$

Where, $B_i$ represents the bandwidth consumption of the $i^{th}$ communication link, $T_i$ denotes the transmission time for data over the $i^{th}$ link, and $n$ is the total number of communication links in the network. In our experiments, to accurately determine the communication cost in our proposed scheme, several key components were considered. First, bandwidth consumption ($B_i$) plays a crucial role, as it represents the amount of data transmitted during interactions between vehicles (V2V) and between vehicles and roadside units (V2RSU). Higher bandwidth usage can lead to increased communication overhead, impacting overall network efficiency. Second, transmission time ($T_i$) was analyzed to measure the duration required for message exchanges over a communication link. Efficient transmission mechanisms help reduce delays and enhance real-time communication. Finally, the number of communication links (n) in the network was taken into account, as a higher number of active communication paths can contribute to increased resource utilization. By incorporating these components into our mathematical model, we were able to quantify communication costs effectively, ensuring an optimized balance between security and performance. According to Fig 6, in the case of V2V communication, our scheme demonstrates notably lower communication costs, with 534 and 530 bits compared to 1664, 2912, and 2560 bits for references [10,19], and [20], respectively. Similarly, for V2RSU communication, our scheme exhibits reduced communication costs, registering values of 530 compared to 1920, 4416, and 2880 in references [10,19], and [20], respectively. These findings underscore the efficiency and optimization achieved in communication resource utilization through our proposed scheme, suggesting its superiority in minimizing communication overhead in VANET systems.

**C. Cache hit ratio.** We evaluate the effective usage of cache while receiving queries from the vehicles. In a VANET system employing a cache-based anonymizer server situated between OBUs and RSUs, the cache hit ratio $R_{hit}$ represents the proportion of queries successfully resolved from the cache compared to the total number of queries made. It can be expressed as:

$$R_{hit} = \frac{N_{hit}}{N_{query}}$$

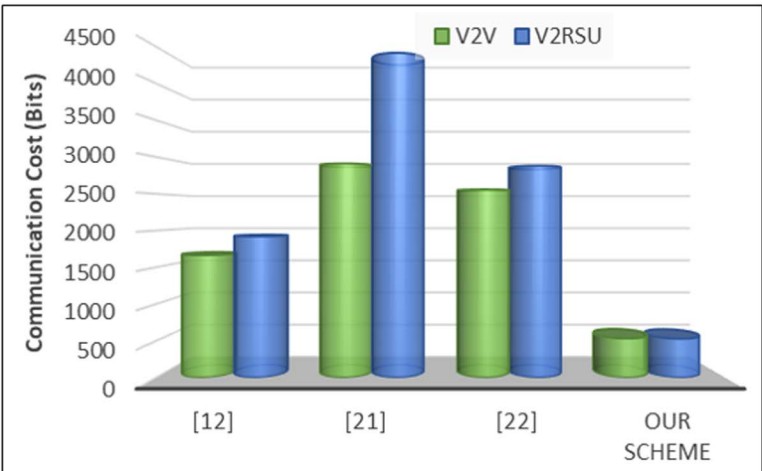

**Fig 6. communication cost.**

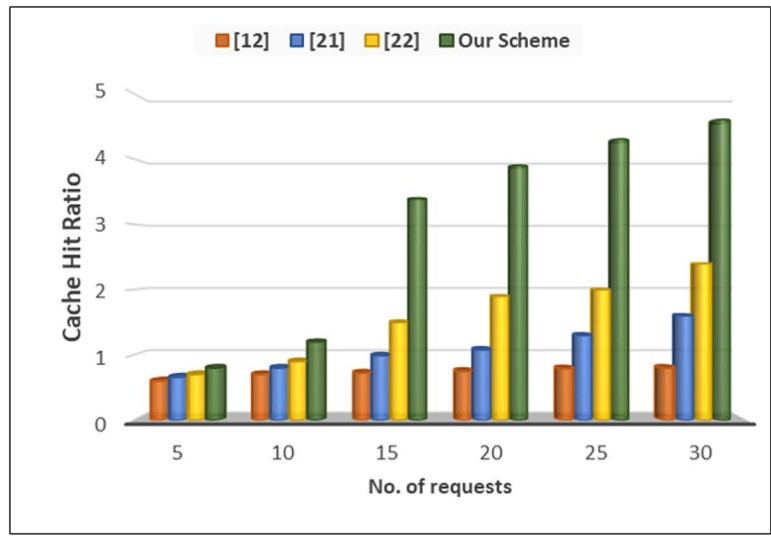

**Fig 7. Cache hit ratio.**

To calculate the cache hit ratio, we considered two key components including number of cache hits ($N_{hit}$) that represents the number of queries that were successfully answered using stored responses in the cache, eliminating the need to forward them to the RSU, and total number of queries ($N_{query}$) which includes all queries received by the anonymizer server from vehicles, whether resolved through caching or forwarded to the RSU. Fig 7 presents the impact of cache usage that significantly improve the overall system. We observe that at initial level, cache hit ratio is almost in all methods, but a significant difference once the number of queries increased.

**D. Location Leakage Probability (L2P).** To calculate the location leakage probability in VANET systems, we define it as the probability that an adversary can accurately infer the location of a vehicle based on the information available to them. Mathematically, it can express as:

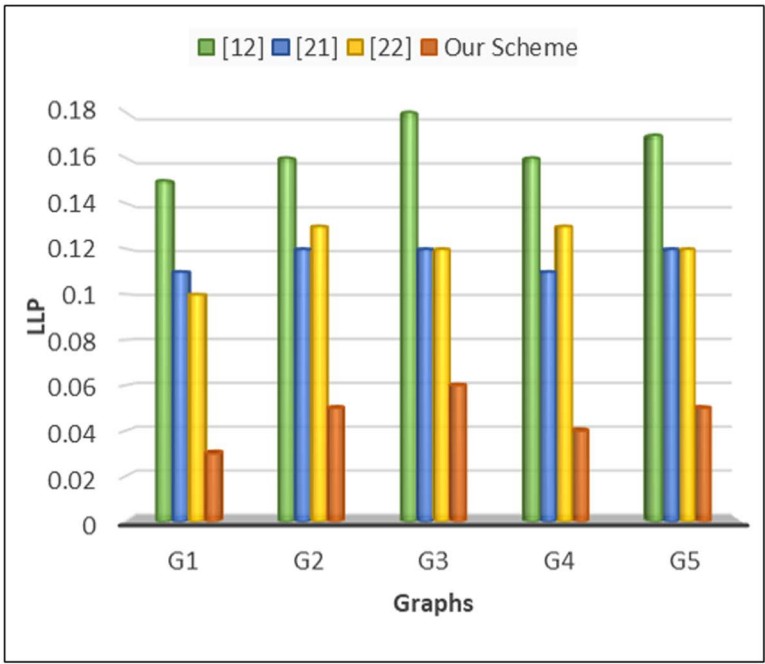

**Fig 8. Location leakage probability.**

$$P \left(Loc\ Leakage\ =\ \frac{No.\ of\ successful\ loc\ interference}{Total\ no.\ of\ requests}\right)$$

Where $P$ (Location Leakage) is the location leakage probability, "Number of Successful Location Inferences" represents the instances where the adversary correctly identifies the location of the vehicle, and "Total Number of Location Queries" denotes the total number of times the adversary attempts to infer the location of the vehicle. This evaluation can be performed over a given period or set of scenarios to assess the effectiveness of privacy protection mechanisms in the VANET system. As depicted in Fig 8, the probability L2P stands at 0.15 with centralized architecture, ranging between 0.1 and 0.12 with distributed architecture, and between 0.1 and 0.13 with cache-based centralized architecture. In contrast, our proposed architecture lowers the probability to approximately 0.05%. Consequently, the effectiveness of L2P protection is significantly enhanced with our proposed architecture.

**E. Entropy.** In the experiments, we evalute our last metric entropy denoted as $H(X)$ that represents the measure of uncertainty or randomness in a system. In the context of VANET systems, entropy is often used to quantify the level of privacy risk associated with the dissemination of data. The calculation of entropy involves the probability distribution of different states or outcomes within the system. Mathematically, entropy is calculated using the formula:

$$H(X) = -\sum_{i=1}^{n} p\left(x_i\right) . \log_2\left(p\left(x_i\right)\right)$$

Where, $H(X)$ is the entropy of the system $X$. $p\left(x_i\right)$ represents the probability of the $i^{th}$ state or outcome occurring. $n$ is the total number of possible states or outcomes in the system. The primary factors contributing to entropy include pseudonymization and anonymization, where vehicle identifiers are replaced with temporary pseudonyms, making it challenging for attackers to track vehicles over time. Additionally, we incorporated cloaking regions, which introduce spatial uncertainty

by ensuring that vehicle locations are not precisely disclosed but rather obscured within a predefined area, increasing randomness in adversarial inference. By analyzing the frequency or likelihood of different information states, such as vehicle locations or identities, entropy provides insights into the level of privacy protection or risk within the system. Higher entropy values indicate greater uncertainty or randomness, while lower values suggest more predictable or deterministic behavior, which may pose privacy concerns in VANET environments. Fig 9 shows that our proposed scheme outperformed the existing methods and achieve maximum entropy level up to 5.

The proposed blockchain-based privacy-preserving scheme for VANETs effectively addresses key security and privacy challenges while maintaining system efficiency. Traditional approaches, such as k-anonymity mechanisms and encryption-based authentication, have demonstrated moderate success in securing VANET communications but often suffer from high computational costs and adaptability issues in dynamic traffic environments. A comparative analysis of our proposed privacy provisioning scheme with existing state-of-the-art methods for user privacy protection in VANETs is presented in Table 3. Our framework overcomes these limitations by integrating a cache-based anonymizer server strategically positioned between On-Board Units (OBUs) and Road-Side Units (RSUs), significantly enhancing privacy while reducing computational and communication overhead. Performance evaluations conducted using the Veins simulation platform demonstrate that our approach outperforms existing schemes in key metrics such as computation cost, communication cost, cache hit ratio, location leakage probability, and entropy analysis. Specifically, our model achieves a computation cost of 0.23972 ms for V2V and 0.13166 ms for V2RSU, significantly lower than previous methods. Additionally, our scheme minimizes communication costs, requiring only 534 and 530 bits for V2V and V2RSU communication, respectively, compared to existing frameworks that demand up to 4416 bits, highlighting superior bandwidth efficiency. The cache hit ratio analysis further underscores the efficiency of our anonymization approach, reducing reliance on centralized servers while maintaining a high success rate in data retrieval. Our framework enhances privacy by reducing location leakage probability to 0.05%, a significant improvement over prior schemes that range between 0.1 to 0.15. Furthermore, entropy evaluation reveals a maximum entropy value of 5, ensuring higher randomness and improved resistance against adversarial inference attacks. The integration of smart contracts and decentralized authentication further strengthens security by mitigating threats such as replay, impersonation, and man-in-the-middle attacks. A comparative analysis with state-of-the-art privacy-preserving schemes confirms that our blockchain-based model achieves the highest privacy rate of

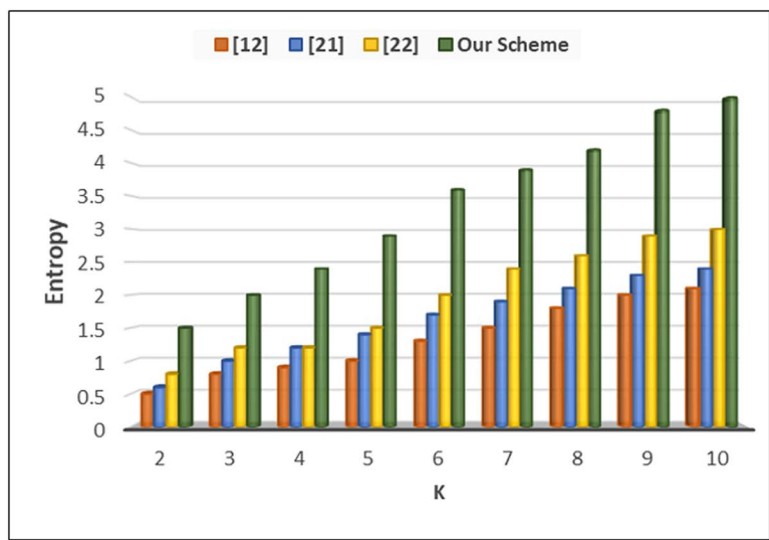

**Fig 9. Anonymous entropy in k-anonymity mechanism.**

**Table 3. Comparative analysis of our proposed scheme with existing privacy provisioning methods for VANETs.**

| Ref. No. | Proposed Model | Used Mechanism | Privacy Rate (%) | Limitations |
|---|---|---|---|---|
| [5] | k-Anonymity Mechanism | Maximum entropy principle to select k vehicles for anonymity | 85% | Limited adaptability in dynamic traffic scenarios |
| [3] | Encryption-Based VANET Authentication | Asymmetric (PKI, ECDSA) and symmetric (group signature) encryption | 90% | Computationally expensive, potential key management issues |
| [12] | Fog-Based Pseudonym Distribution | Fog computing for location privacy | 80% | Vulnerable to fog node attacks, potential processing delays |
| [13] | Trajectory Privacy with Mix Zones | Multiple mix zones for unlinkability of pseudonyms | 88% | May not be effective in sparsely populated areas |
| [14] | Blocktree | Blockchain-based signature embedding | 92% | High storage requirements and latency |
| [15] | Lightweight Blockchain for VANET | Scalable blockchain with decentralized privacy protection | 91% | Scalability issues in large-scale networks |
| [16] | Blockchain-Based Key Management | Distributed key management with blockchain | 90% | Complexity in heterogeneous vehicle networks |
| [17] | Blockchain-Based Proof of Location | Privacy-preserving location verification | 89% | Potential latency in proof validation |
| [18] | Smart Contract-Based Key Agreement | Decentralized key agreement using smart contracts and Bloom filters | 87% | Bloom filters introduce additional latency, prone to MITM attacks |
| [19] | Batch Authentication for IoV | Blockchain-based V2V and V2RSU authentication using pBFT | 91% | Cloud server increases system complexity and latency |
| [20] | Vehicle-to-Infrastructure Authentication | Blockchain-based RSU authentication and pseudo-identity storage | 90% | Lacks V2V authentication, limiting overall security scope |
| [21] | TCEMD Trust Model | Trust-cascading mechanism integrating direct and indirect trust | 89% | Susceptible to manipulation via indirect trust, computational overhead |
| [22] | TROVE (Context-Aware Trust Model) | Reinforcement learning-based trust evaluation | 92% | Computationally intensive, adversarial RL exploits, scalability challenges |
| [23] | SRE (Secure Reputation Enhancement) | ECC and Paillier encryption-based reputation system | 93% | Semi-trusted CSP poses security risks, cloud latency issues |
| **Our Scheme** | **blockchain-based decentralized framework** | **cache based anonymizer, lightweight consortium blockchain** | **96.07%** | **Used OPNET simulation. Real environment implementation required.** |

96.07%, surpassing existing solutions while maintaining low operational costs and high efficiency. Despite these advancements, real-world deployment and scalability assessments remain critical areas for future exploration. These findings emphasize the potential of our proposed framework to revolutionize secure and efficient VANET communications, making it a promising solution for real-world intelligent transportation systems.

## 7. Conclusion and future directions

The proposed blockchain based privacy provisioning scheme for VANET systems introduces a multi-layered approach aimed at protecting sensitive information against potential threats. By incorporating various privacy protection mechanisms at different stages, the system ensures robust privacy provisioning throughout. Initially, anonymizer servers (AS) anonymize vehicle location and identity information, enhancing security during data transmission to RSUs via OBUs. This anonymization process, coupled with the incorporation of cloaking regions, establishes a secure pathway at the outset. Subsequently, within the core network, Certificate Authorities (CAs) generate certificates to authenticate queries received, adding another layer of protection. Moreover, the deployment of a private chain within the blockchain network ensures secure storage of hash values, thwarting tampering attempts effectively. Smart contracts further fortify the system

by automating predefined rules and conditions, minimizing human intervention while enhancing accuracy and efficiency. Agent nodes, particularly miners, play a vital role in maintaining the integrity of transactions through consensus mechanisms, contributing to the system's robustness. The proposed solution demonstrates resilience against various attacks, including query content modification, content replay, impersonation, and middle-man attacks, as evidenced by mathematical formulations and proofs. Through simulation experiments, the effectiveness of the proposed architecture is evaluated against centralized and decentralized approaches, highlighting significant improvements in computation cost, communication efficiency, cache hit ratio, location leakage probability, and entropy. The results demonstrate the superiority of the proposed scheme in minimizing computational overhead, optimizing resource utilization, enhancing privacy protection, and maximizing entropy levels, making it a promising solution for ensuring privacy in VANET systems.

Though in this research, our scheme outperforms the existing state of the art studies, however, by future perspectives, our aim is to implement the proposed scheme in real environment to know the effectives in real scenarios.

## Author contributions

**Conceptualization:** M. Usman Ashraf, Fatmah Alanazi.

**Data curation:** Iqra Ilyas.

**Formal analysis:** Iqra Ilyas, M. Usman Ashraf, Khlood Shinan.

**Funding acquisition:** Sami S. Albouq.

**Investigation:** Iqra Ilyas, Sami S. Albouq, Hanan E. Alhazmi, Saqib Ali.

**Methodology:** M. Usman Ashraf, Hanan E. Alhazmi.

**Project administration:** Hanan E. Alhazmi, Saqib Ali.

**Resources:** Sami S. Albouq.

**Software:** Fatmah Alanazi.

**Validation:** Iqra Ilyas, Sami S. Albouq, Khlood Shinan, Saqib Ali.

**Visualization:** Iqra Ilyas.

**Writing – original draft:** M. Usman Ashraf.

**Writing – review & editing:** Iqra Ilyas, Khlood Shinan, Hanan E. Alhazmi, Fatmah Alanazi, Saqib Ali.

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
