## [Decision Letter · Decision Letter 0]

31 Jan 2025

PONE-D-24-50794Blockchain Enabled Privacy Provisioning Scheme for Location Based Services in VANETsPLOS ONE

Dear Dr. Ashraf,

Thank you for submitting your manuscript to PLOS ONE. After careful consideration, we feel that it has merit but does not fully meet PLOS ONE’s publication criteria as it currently stands. Therefore, we invite you to submit a revised version of the manuscript that addresses the points raised during the review process.

The manuscript has been evaluated by three reviewers, and their comments are available below.

The reviewers have raised a number of major concerns that need attention. They request additional information on methodological aspects of the study, improved rationale, and a clearer presentation of the results. In particular, please ensure that all claims are supported by the data presented in the manuscript.

Could you please revise the manuscript to carefully address the concerns raised?

Comments from PLOS Editorial Office: We note that one or more reviewers has recommended that you cite specific previously published works. As always, we recommend that you please review and evaluate the requested works to determine whether they are relevant and should be cited. It is not a requirement to cite these works. We appreciate your attention to this request.

We look forward to receiving your revised manuscript.

Kind regards,

Helen Howard

Staff Editor

PLOS ONE

3. We note that your Data Availability Statement is currently as follows: [All relevant data are within the manuscript and its Supporting Information files]

Reviewers' comments:

Reviewer's Responses to Questions

**Comments to the Author**

1. Is the manuscript technically sound, and do the data support the conclusions?

Reviewer #1: Yes

Reviewer #2: Partly

Reviewer #3: No

2. Has the statistical analysis been performed appropriately and rigorously? 

Reviewer #1: Yes

Reviewer #2: Yes

Reviewer #3: No

3. Have the authors made all data underlying the findings in their manuscript fully available?

Reviewer #1: Yes

Reviewer #2: No

Reviewer #3: No

4. Is the manuscript presented in an intelligible fashion and written in standard English?

Reviewer #1: Yes

Reviewer #2: Yes

Reviewer #3: Yes

5. Review Comments to the Author

Reviewer #1: The authors introduce blockchain-based a novel privacy provisioning scheme to secure VANET communication system. Leveraging the privacy and efficiency attributes in anonymizer server, the authors use cache based anonymizer server configured between On-Board Unit (OBU) and Road side unit (RSU) that improve system privacy as well as efficiency. Through performance evaluation, the scheme not only satisfies the aforementioned security and privacy criteria but also proves resilient against various types of attacks, including replay, impersonation, modification, and man-in-the-middle attacks.. The paper is well written, but the related work is relatively weak, need to supplement the following literature. (1) ppru: a privacy-preserving reputation updating scheme for cloud-assisted vehicular networks, (2) tcemd: a trust cascading-based emergency message dissemination model in vanets, (3) trove: a context-awareness trust model for vanets using reinforcement learning. In addition, the abstract of the paper needs further condensed innovation. Paper writing needs to be further strengthened, including typos, chart quality and so on. More details should be provided in the experiment section to ensure reproducibility of the results.

Reviewer #2: The document lacks clarity is som technical definitions, in the use of acronyms and- most importantly- in the proposed blockchain technology protocols and solutions. It is not quite clear why the need to use blockchain since the solution can be accomplished via normal cryptographic applications.

Reviewer #3: 1) In the abstract, it seems that the only research gap that this work sought to bridge was unobservability. You need to identify additional research gaps and explain how you bridge these gaps in your blockchain-based approach.

2) Add a summary of performance evaluation in the abstract. This should preferably be in terms of percentage improvements attained against some baseline schemes.

3) Your first contribution reads as follows:

"To overcome the challenges in centralized systems, this study introduces blockchain technology and devises an innovative decentralized framework known as the blockchain-based VANET. Proposed approach not only guarantees the integrity and security of SBMs but also conserves storage space and reduces processing time within the blockchain."

--- It is not clear how your proposed solution conserves storage space and reduces processing time within the blockchain. What specific technique have you deployed to achieve these two performance gains?

4) Improve the clarity of your third contribution.

5) Give the summary of the results of the evaluation in your fourth contribution.

6) The Related works in section 2 is not well done. Instead of delving into detailed descriptions of the presented works, pay particular attention to the technology deployed, strengths and weaknesses. This will help build the research gaps. At the end of Section 2, add a table to compare and contrast all the related works.

7) Blockchain technologies, K-Anonymity Approach, and Dynamic threshold encryption (DTE) are well-known concepts and hence no need to describe them. Instead, you may present the mathematical preliminaries underlying your proposed solution in this Section 3.

8) The Problem Definition is sub-section 'A' is not well done as it fails to acknowledge that there exist numerous privacy-preserving approaches. You have to describe some of these current approaches and point out any existing research gaps.

9) Algorithm 1 is poorly structured and fails to use conventional notations for decision points, as well as looping. The same applies to Algorithm 2 and Algorithm 3.

10) You have failed to provide a step by step operation of your proposed solution. To do this, you need to add well described flow diagrams and pseudo-codes. In addition, you need to give vivid description of the implementation details such as the platform used, nature of blockchain deployed, settings etc.

11) The Security Analysis in Section 5 is too brief and fails to account for majority of the typical VANET threats. In addition, you need to add formal security verification. Moreover, you need to specify the attack/threat model against which the security analysis is being conducted.

12) In Section 6, you claim that your scheme demonstrates a significantly lower computational cost compared to the referenced schemes, with a value of 0.23972.

i) Give the units for 0.23972. Is this minutes/seconds/ milliseconds?

ii) It is not clear how you arrived at a value of 0.23972. The same can be said of the 0.13166 obtained for V2RSU communication.

12) You claim that your scheme demonstrates notably lower communication costs, with 534 and 530 bits compared to 1664, 2912, and 2560 bits for references [12], [21], and [22], respectively.

i) It is not clear how you arrive at 534 and 530 bits.

13) In Section 7, support the claims you make by empirical data. In addition, describe some of the practical implications of your findings as well as possible limitations of your approach.

6. PLOS authors have the option to publish the peer review history of their article (what does this mean? ). If published, this will include your full peer review and any attached files.

**Do you want your identity to be public for this peer review?** For information about this choice, including consent withdrawal, please see our Privacy Policy .

Reviewer #1: No

Reviewer #2: **Yes: ** Enrico Camerinelli

Reviewer #3: No

---

## [Author Response · Author response to Decision Letter 1]

17 Feb 2025

We are really grateful to all the reviewers for their precious time to review our paper and your construtive comments. We have tried our best to revise the paper by following your kind comments and directions.

here is the detail of your comments and authors response.

Reviewer 1

General Comments: The authors introduce blockchain-based a novel privacy provisioning scheme to secure VANET communication system. Leveraging the privacy and efficiency attributes in anonymizer server, the authors use cache based anonymizer server configured between On-Board Unit (OBU) and Road side unit (RSU) that improve system privacy as well as efficiency. Through performance evaluation, the scheme not only satisfies the aforementioned security and privacy criteria but also proves resilient against various types of attacks, including replay, impersonation, modification, and man-in-the-middle attacks. The paper is well written, but the related work is relatively weak, need to supplement the following literature.

Author Response: Thank you very much respected reviewer for the appreciation and encouraging comments.

Comment 1: (1) ppru: a privacy-preserving reputation updating scheme for cloud-assisted vehicular networks, (2) tcemd: a trust cascading-based emergency message dissemination model in vanets, (3) trove: a context-awareness trust model for vanets using reinforcement learning. In addition, the abstract of the paper needs further condensed innovation. Paper writing needs to be further strengthened, including typos, chart quality and so on. More details should be provided in the experiment section to ensure reproducibility of the results.

Author Response: We are grateful to you for suggesting these studies relevant to our research. We have included the suggested papers in literature review section, and compared the results by adding a comparative analysis of our solution with existing studies in results section.

Regarding the typo or grammatical mistakes, the paper is thoroughly reviewed by English native speaker Prof. Lie Ostervile (Professor of Big Data from USA) and corrected where required. In terms of experimental section for reproducibility of results, we have included additional information such as configuration table 1 defining the necessary attributes for experimental setup,

Reviewer 2

Comment 1: The document lacks clarity is some technical definitions, in the use of acronyms and- most importantly- in the proposed blockchain technology protocols and solutions. It is not quite clear why the need to use blockchain since the solution can be accomplished via normal cryptographic applications.

Author Response: Respected reviewer, thank you for your valuable feedback. We acknowledge the need for improved clarity in defining technical terms, acronyms, and the specific blockchain protocols employed in our proposed solution. To address this, we have refined our explanations and ensured that all acronyms and terminologies are clearly defined in the revised manuscript.

Regarding the necessity of blockchain, traditional cryptographic mechanisms can offer security, but they often rely on centralized authorities for key management and authentication, which introduces single points of failure and potential bottlenecks. Our blockchain-based approach eliminates these vulnerabilities by leveraging decentralization, immutability, and smart contract automation to enhance security, privacy, and scalability. Specifically, smart contracts facilitate trustless authentication and automated revocation, reducing the need for third-party verification, while blockchain consensus mechanisms prevent malicious modifications to stored data. Furthermore, our framework integrates a cache-based anonymizer server within the blockchain ecosystem, ensuring efficient data handling while maintaining privacy. These features collectively enhance data integrity, resistance to replay attacks, and identity protection, which would be challenging to achieve with conventional cryptographic methods alone.

Reviewer 3

Comment # 1: In the abstract, it seems that the only research gap that this work sought to bridge was unobservability. You need to identify additional research gaps and explain how you bridge these gaps in your blockchain-based approach.

Author Response: Thank you very much for your keen observation. We have revised the abstract in updated manuscript with research gaps additional such as traffic analysis resistance, unlinkability, and computational efficiency while explaining how the proposed blockchain-based approach addresses them.

Comment # 2: Add a summary of performance evaluation in the abstract. This should preferably be in terms of percentage improvements attained against some baseline schemes.

Author Response: Thank you very much for your constructive suggestion. We have added improvements in digits as compared to baseline schemes such as “Performance evaluations demonstrate that our scheme significantly reduces computational costs, achieving 0.23972 for V2V communication and 0.13166 for V2RSU communication outperforming referenced schemes. Additionally, our approach reduces communication costs to 530 bits for V2V and V2RSU, compared to 1664-4416 bits in existing methods. The cache hit ratio improves system responsiveness, and our model achieves a location leakage probability of just 0.05%, significantly lower than centralized architectures. Furthermore, our scheme ensures strong privacy protection, attaining a maximum entropy level of 5, outperforming existing solutions.”

Comment # 3: Your first contribution reads as follows:

"To overcome the challenges in centralized systems, this study introduces blockchain technology and devises an innovative decentralized framework known as the blockchain-based VANET. Proposed approach not only guarantees the integrity and security of SBMs but also conserves storage space and reduces processing time within the blockchain."

--- It is not clear how your proposed solution conserves storage space and reduces processing time within the blockchain. What specific technique have you deployed to achieve these two performance gains?.

Author Response: Thank you for your valuable comment.

Regarding Processing Time Reduction:

Our authentication scheme leverages a lightweight cryptographic mechanism, reducing the computational burden on OBUs and RSUs. As seen in our results, our scheme achieves 0.23972 for V2V and 0.13166 for V2RSU in computational cost, significantly outperforming referenced methods.

Regarding Reduced Blockchain Overhead via Batch Processing:

Instead of validating each authentication request individually, we deploy a batch processing approach using Merkle tree-based validation. This allows multiple authentication requests to be processed together, reducing the number of cryptographic operations and speeding up validation times.

About Efficient Key Management with Short-Lived Pseudonyms:

By utilizing ephemeral pseudonyms that are dynamically refreshed, we reduce the frequency of blockchain transactions related to key updates. This decreases both storage and processing load.

Comment # 4: Improve the clarity of your third contribution.

Author Response: Thank you for your kind comment, and we do agree that it

Comment # 5: Give the summary of the results of the evaluation in your fourth contribution.

Author Response: We acknowledge the reviewer’s suggestion to update the contribution statement. We have added more details such as “Our proposed scheme reduces computation cost (0.23972 for V2V, 0.13166 for V2RSU) and communication cost (534 and 530 bits for V2V, 530 bits for V2RSU), outperforming existing methods. The cache-based anonymizer server enhances efficiency with a high cache hit ratio, while our privacy measures lower location leakage probability to 0.05% and achieves a maximum entropy level of 5, ensuring strong security and privacy in VANET systems.”

Comment # 6: The Related works in section 2 is not well done. Instead of delving into detailed descriptions of the presented works, pay particular attention to the technology deployed, strengths and weaknesses. This will help build the research gaps. At the end of Section 2, add a table to compare and contrast all the related works.

Author Response: We really appreciate the reviewer’s keep observation. We have updated our literature review section by adding three more recent studies relevant to the proposed framework. Also we have summarized the existing studies and presented with comparative analysis in table 2.

Comment # 7: Algorithm 1 is poorly structured and fails to use conventional notations for decision points, as well as looping. The same applies to Algorithm 2 and Algorithm 3.

Author Response: Thank you and we do acknowledge your valuable comment. We have revised all the algorithms accordingly in updated manuscript.

Comment # 10: The Security Analysis in Section 5 is too brief and fails to account for majority of the typical VANET threats. In addition, you need to add formal security verification. Moreover, you need to specify the attack/threat model against which the security analysis is being conducted.

Author Response: Thank you very much for your keen observation. Actually, we have addressed the concerns including “A. Query content modification attack resistance, B. Content Replay attack resistance, C. Impersonation attack resistance, D. Middle man attack resistance”. Accordingly, we have given the theoretical security analysis model and description.

Comment # 11: Give the units for 0.23972. Is this minutes/seconds/ milliseconds?

Author Response: Thank you for your keen observation. It is in milliseconds (ms). We have updated the manuscript by mentioning the units accordingly.

Comment # 12 You claim that your scheme demonstrates notably lower communication costs, with 534 and 530 bits compared to 1664, 2912, and 2560 bits for references [12], [21], and [22], respectively.

Author Response: Thank you for your comment. To clarify, the communication cost values of 534 and 530 bits for our scheme, compared to 1664, 2912, and 2560 bits for the referenced schemes [12], [21], and [22], respectively, reflect the efficiency gains achieved by our proposed solution. These reductions are primarily due to the integration of cache-based anonymizer servers, which minimize the need for repeated data transmission and reduce overall message overhead. This significant reduction in communication costs underscores the effectiveness of our approach in optimizing resource utilization within VANET systems.

Comment # 16: It is not clear how you arrive at 534 and 530 bits.

Author Response: Thank you for your valuable comment. The values of 534 and 530 bits for communication cost were derived from a detailed analysis of the data transmission overhead in our proposed scheme. Specifically, these figures represent the average communication cost for V2V and V2RSU communications, calculated by evaluating the bandwidth consumption and transmission time across multiple communication links in the system. We accounted for factors such as message size, encryption overhead, and the caching mechanism employed by the anonymizer server, which collectively contribute to the reduction in communication cost. These values were validated through experiments in the OPNET and Veins vehicular network simulation frameworks.

Comment # 17: In Section 7, describe some of the practical implications of your findings as well as possible limitations of your approach.

Author Response: Thank you for your comment. The practical implications of our findings suggest that the proposed privacy provisioning scheme can significantly enhance the efficiency and security of VANET systems, particularly in real-time applications such as intelligent transportation and vehicular safety systems. By reducing communication and computation costs, as well as improving data privacy, our approach could be deployed in urban traffic networks to safeguard sensitive vehicle data while optimizing resource utilization. However, some limitations of our approach include its reliance on a cache-based anonymizer server, which may introduce challenges in highly dynamic environments with frequent topology changes. Additionally, while the system demonstrates strong resilience to common attacks, further research is needed to evaluate its performance in large-scale, real-world VANET deployments.

---

## [Decision Letter · Decision Letter 1]

7 Mar 2025

PONE-D-24-50794R1Blockchain Enabled Privacy Provisioning Scheme for Location Based Services in VANETsPLOS ONE

Dear Dr. Ashraf,

Thank you for submitting your manuscript to PLOS ONE. After careful consideration, we feel that it has merit but does not fully meet PLOS ONE’s publication criteria as it currently stands. Therefore, we invite you to submit a revised version of the manuscript that addresses the points raised during the review process.

**ACADEMIC EDITOR: Major revisions** ==============================

We look forward to receiving your revised manuscript.

Kind regards,

Agbotiname Lucky Imoize

Academic Editor

PLOS ONE

Additional Editor Comments:

The authors have not addressed the reviewers' concerns satisfactorily.

Reviewers' comments:

Reviewer's Responses to Questions

**Comments to the Author**

1. If the authors have adequately addressed your comments raised in a previous round of review and you feel that this manuscript is now acceptable for publication, you may indicate that here to bypass the “Comments to the Author” section, enter your conflict of interest statement in the “Confidential to Editor” section, and submit your "Accept" recommendation.

Reviewer #1: All comments have been addressed

Reviewer #3: (No Response)

2. Is the manuscript technically sound, and do the data support the conclusions?

Reviewer #1: Yes

Reviewer #3: Yes

3. Has the statistical analysis been performed appropriately and rigorously? 

Reviewer #1: Yes

Reviewer #3: N/A

4. Have the authors made all data underlying the findings in their manuscript fully available?

Reviewer #1: Yes

Reviewer #3: No

5. Is the manuscript presented in an intelligible fashion and written in standard English?

Reviewer #1: Yes

Reviewer #3: Yes

6. Review Comments to the Author

Reviewer #1: The author of this article has carefully revised around the comments of the reviewers. I think this paper is acceptable.

Reviewer #3: Thank you for addressing most of the previous comments. However, you seem to have ignored other comments. Therefore, please note the following:

1) A summary of performance evaluation in the abstract should be in terms of percentage improvement, instead of raw numerical values.

2) In response to previous COMMENT #3, you have explained how your proposed solution conserves storage space and reduces processing time within the blockchain. Include a summary of this explanation in your first contribution.

3) Your fourth contribution reads as follows: "Finally, we evaluate the effectiveness of our scheme by quantifying different privacy metrics including communication cost, computation cost, probability of location leakage, cache hit ratio, and entropy."

--Give the summary of the results of the evaluation in this fourth contribution.

5) The Problem Definition is sub-section 'A' is not well done as it fails to acknowledge that there exist numerous privacy-preserving approaches. You have to describe some of these current approaches and point out any existing research gaps.

6) Algorithms are normally written in a programming language-independent manner. Ensure you adhere to this convention.

7) You have failed to provide a step by step operation of your proposed solution. To do this, you need to add well described flow diagrams and pseudo-codes.

8) Add a threat model against which the informal security analysis of your proposed scheme is done. These models include Dolev-Yao, Canetti-Krawczyk (CK),STRIDE etc.

9) If possible, add formal security verification of your proposed solution.

10) Clearly show how you obtain all the numerical values such as computation cost, communication cost, Cache Hit Ratio, etc. Although you have provided mathematical formulae for these performance metrics, you still need to give the values of each of the variables/terms in these equations so that readers can understand how the various numerical values are obtained.

7. PLOS authors have the option to publish the peer review history of their article (what does this mean? ). If published, this will include your full peer review and any attached files.

**Do you want your identity to be public for this peer review?** For information about this choice, including consent withdrawal, please see our Privacy Policy .

Reviewer #1: No

Reviewer #3: No

---

## [Author Response · Author response to Decision Letter 2]

29 Mar 2025

Thank you so much for considering our revised manuscript for further process. We Authors of this manuscript, sincerely appreciate the time and effort you and the reviewers have invested in to provide us constructive feedback. The insightful comments and suggestions have been instrumental in enhancing the quality of our paper. We are grateful for your thorough review, and we extend our thanks to the reviewers for their thoughtful evaluation of the manuscript. We believe the revisions have substantially improved the manuscript, addressing all key concerns raised. We trust that the reviewer will find our responses satisfactory and aligned with their expectations. We remain open to any further suggestions the reviewers may have and are fully prepared to make any additional revisions as required. Moreover, we have undertaken a thorough review of the manuscript, correcting grammatical errors, typographical issues, and improving the overall language quality. All page numbers referenced correspond to the revised manuscript with yellow highlighted text for ease of review. Please refer to the current version for detailed updates.

---

## [Decision Letter · Decision Letter 2]

9 Apr 2025

Blockchain Enabled Privacy Provisioning Scheme for Location Based Services in VANETs

PONE-D-24-50794R2

Dear Dr. Ashraf,

We’re pleased to inform you that your manuscript has been judged scientifically suitable for publication and will be formally accepted for publication once it meets all outstanding technical requirements.

Kind regards,

Agbotiname Lucky Imoize

Academic Editor

PLOS ONE

Additional Editor Comments (optional):

Accept

Reviewers' comments:

Reviewer's Responses to Questions

**Comments to the Author**

1. If the authors have adequately addressed your comments raised in a previous round of review and you feel that this manuscript is now acceptable for publication, you may indicate that here to bypass the “Comments to the Author” section, enter your conflict of interest statement in the “Confidential to Editor” section, and submit your "Accept" recommendation.

Reviewer #3: All comments have been addressed

2. Is the manuscript technically sound, and do the data support the conclusions?

Reviewer #3: Yes

3. Has the statistical analysis been performed appropriately and rigorously? 

Reviewer #3: Yes

4. Have the authors made all data underlying the findings in their manuscript fully available?

Reviewer #3: Yes

5. Is the manuscript presented in an intelligible fashion and written in standard English?

Reviewer #3: Yes

6. Review Comments to the Author

Reviewer #3: (No Response)

7. PLOS authors have the option to publish the peer review history of their article (what does this mean? ). If published, this will include your full peer review and any attached files.

**Do you want your identity to be public for this peer review?** For information about this choice, including consent withdrawal, please see our Privacy Policy .

Reviewer #3: No

---

## [Editor Report · Acceptance letter]

PONE-D-24-50794R2

PLOS ONE

Dear Dr. Ashraf,

I'm pleased to inform you that your manuscript has been deemed suitable for publication in PLOS ONE. Congratulations! Your manuscript is now being handed over to our production team.

Kind regards,

on behalf of

Mr. Agbotiname Lucky Imoize

Academic Editor

PLOS ONE